# Regional connectivity drove bidirectional transmission of SARS-CoV-2 in the Middle East during travel restrictions

Edyth Parker [1,15] ✉, Catelyn Anderson[1,15], Mark Zeller[1,15], Ahmad Tibi[2], Jennifer L. Havens [3], Geneviève Laroche[4], Mehdi Benlarbi[4], Ardeshir Ariana[4], Refugio Robles-Sikisaka[1], Alaa Abdel Latif[1], Alexander Watts[5], Abdalla Awidi[6,7], Saied A. Jaradat[8], Karthik Gangavarapu[9], Karthik Ramesh[1], Ezra Kurzban[1], Nathaniel L. Matteson[1], Alvin X. Han [10], Laura D. Hughes [11], Michelle McGraw[1], Emily Spencer[12], Laura Nicholson [12], Kamran Khan[5], Marc A. Suchard [9,13], Joel O. Wertheim[14], Shirlee Wohl[1], Marceline Côté[4], Amid Abdelnour[2], Kristian G. Andersen [1,16] & Issa Abu-Dayyeh [2,6,16] ✉

Regional connectivity and land travel have been identified as important drivers of SARS-CoV-2 transmission. However, the generalizability of this finding is understudied outside of well-sampled, highly connected regions. In this study, we investigated the relative contributions of regional and intercontinental connectivity to the source-sink dynamics of SARS-CoV-2 for Jordan and the Middle East. By integrating genomic, epidemiological and travel data we show that the source of introductions into Jordan was dynamic across 2020, shifting from intercontinental seeding in the early pandemic to more regional seeding for the travel restrictions period. We show that land travel, particularly freight transport, drove introduction risk during the travel restrictions period. High regional connectivity and land travel also drove Jordan's export risk. Our findings emphasize regional connectedness and land travel as drivers of transmission in the Middle East.

Global pathogen surveillance programs have generated an unprecedented number of SARS-CoV-2 sequences since the start of the COVID-19 pandemic, enabling large- and fine-scale characterization of transmission dynamics[1–4]. Several studies have highlighted regional and neighboring countries as major sources of transmission, with land travel identified as an important driver of viral spread[2,5–8]. However, the generalizability of these findings remains understudied in regions with sparser sampling and more restricted freedom of movement than

---

[1]Department of Immunology and Microbiology, The Scripps Research Institute, La Jolla, CA, USA. [2]Biolab Diagnostic Laboratories, Amman, Jordan. [3]Bioinformatics and Systems Biology Graduate Program, University of California San Diego, La Jolla, CA, USA. [4]Department of Biochemistry, Microbiology and Immunology, and Center for Infection, Immunity, and Inflammation, University of Ottawa, Ottawa, ON, Canada. [5]Bluedot, Toronto, ON, Canada. [6]Cell Therapy Center, The University of Jordan, Amman, Jordan. [7]Thrombosis, haemostasis laboratory, School of Medicine, The University of Jordan, Amman, Jordan. [8]Princess Haya Biotechnology Center, Jordan University of Science and Technology, Irbid, Jordan. [9]Department of Human Genetics, David Geffen School of Medicine, University of California, Los Angeles, Los Angeles, CA, USA. [10]Department of Medical Microbiology & Infection Prevention, Amsterdam UMC, University of Amsterdam, Amsterdam Institute for Infection and Immunity, Amsterdam, the Netherlands. [11]Department of Integrative, Structural and Computational Biology, The Scripps Research Institute, La Jolla, CA 92037, USA. [12]Scripps Research Translational Institute, La Jolla, CA, USA. [13]Department of Biostatistics, Fielding School of Public Health, University of California, Los Angeles, Los Angeles, CA, USA. [14]Department of Medicine, University of California San Diego, La Jolla, CA, USA. [15]These authors contributed equally: Edyth Parker, Catelyn Anderson, Mark Zeller. [16]These authors jointly supervised this work: Kristian G. Andersen, Issa Abu-Dayyeh. ✉e-mail: eparker@scripps.edu; i.abudayyeh@biolab.jo

Europe. It is not clear what the relative contributions of regional and intercontinental connectivity are to the source-sink dynamics for most regions. It is also unclear how these relationships change over time in response to the unprecedented restrictions on human mobility from travel restrictions and comprehensive curfews during the pandemic. It is important to understand how the sources of transmission change over time and in response to mobility restrictions in order to evaluate the effectiveness of interventions such as border closures.

Jordan is in the northern Arabian Peninsula, bordered by Syria, Iraq, Saudi Arabia, Palestine and Israel. Jordan has strong sociopolitical and economic ties with other countries in the Middle East and North Africa region (here referred to as the Middle East). The Middle East accounted for ~60% of in-bound air travel to Jordan in the pre-pandemic months in addition to high volumes of private and commercial land travel. Jordan banned all non-essential travel across its land, air and maritime borders from 17 March 2020 to early September 2020. Border closures were part of a stringent package of non-pharmaceutical interventions (NPIs) implemented in response to an increase in SARS-CoV-2 cases in returning travelers and their direct contacts after the virus was first detected in the country on 2 March 2020. Notably, Jordan implemented a 14-day hotel-quarantine in mid-March, which was only lifted in September 2020[9]. Travel volume declined by 96% in response to the restrictions. During this period, Jordan largely maintained essential travel and commercial transport with only Middle Eastern countries. This suggests a potential shift in transmission to and from Jordan (bidirectional transmission) from global to more regional source-sink dynamics during this period. However, the contribution of regional connectivity to viral spread is currently largely obscured by undersampling in the Middle East. There are only a select number of studies investigating the genomic epidemiology of SARS-CoV-2 in Middle Eastern countries, with limited temporal and geographic scope[10–16].

In this study, we characterize the genomic epidemiology of SARS-CoV-2 in Jordan from March 2020 to the end of its second wave (May 2021). We focus on Jordan's connectivity to the Middle East region as a driver of its transmission dynamics. Toward this, we reconstruct the timing and origin/destination of bidirectional transmission for Jordan by integrating phylogeographic reconstructions, travel data, and an incidence-informed introduction and export intensity index. We assess the robustness of our inferred source-sink dynamics over time across three different downsampling strategies developed to account for sparse sampling particularly in the Middle East. We show that the profile of viral introductions was dynamic during the pandemic, shifting from air travel-driven risk from Europe to regional risk mediated by land freight transport during the period of travel restrictions. We also show that connectivity to the Middle East disproportionately drove Jordan's export risk, with significant contribution from land travel. This was not evident in the genomic data due to a lack of sampling. Our findings emphasize the need for strategies aiming to stop or slow the spread of viral introductions (including new variants) with travel restrictions to prioritize management of risk from regional land travel alongside air travel restrictions.

## Results

### Early introductions into Jordan were predominantly sourced from Europe and successfully contained by NPIs

We investigated the major sources of introductions into Jordan and the shifts in the introduction profile over time, with a focus on the relative contribution of Middle Eastern (defined in Methods) and non-Middle Eastern countries. Toward this, we generated 579 sequences from 16 March 2020 to 31 December 2020 (encompassing the epidemic containment phase and the first epidemic wave) randomly sampled from routine diagnostic tests performed by Biolab Diagnostic Laboratories in five governorates (Fig. 1A, B). Our sampling was predominantly concentrated in the densely populated Amman and Irbid, which were the epicenters of the epidemic across both waves (Fig. 1D).

To determine the profile of introductions into Jordan, we reconstructed the timing and pattern of geographic transitions into and out of Jordan across the full posterior of Bayesian phylogeographic reconstructions. This approach accounts for uncertainty in the phylogenies and migration histories associated with low sequence variability and sampling biases[17]. We assessed the robustness of our findings across three downsampling strategies developed to minimize the effect of sampling biases, particularly undersampling in the Middle East. Of all Middle Eastern countries, only Israel and Qatar sampled >1% of cumulative cases up to October 2021. Notably, Syria and Yemen had no publicly available sequences, with fewer than 150 respective sequences from Libya, Algeria, Lebanon, Iraq, Tunisia, and Palestine during the study period (Fig. 1E). Our downsampling strategies included an epidemic incidence-proportional downsampling (EII), an incidence-informed strategy enriching for Middle Eastern sequences (RE) and an importation intensity-proportional downsampling (III) (see Methods).

Our phylogeographic reconstructions demonstrated that the likely source of introductions into Jordan was dynamic over the course of the pandemic, with early introductions predominantly sourced from Europe (Fig. 2A–C) This is consistent with Europe's high incidence at the time (Supplementary Fig. 4) but also early pandemic sampling biases[1]. Overall, we estimated a lower bound of 28 [posterior median, 95% HPD 25–31] independent introductions into Jordan for the sampled genomic data, with estimates robust to downsampling and randomization (Fig. 2B). We estimated that approximately half of the sampled introductions (14 [posterior median, 95% HPD 12–16] occurred before the travel restrictions were imposed on the 17th of March, with the majority of sampled introductions occurring in March (Fig. 2C). This was largely driven by the high number of singletons sampled during this period, with limited onward transmission at the time (Fig. 2A).

To understand the lineage diversity across Jordan's waves, we assigned lineages to the sequences using Pangolin[18]. We sampled a higher number of distinct lineages in the early pandemic months. This supports the above evidence of multiple independent introductions of distinct lineages in the early stages with no one lineage dominating, indicating limited onward transmission or establishment of these lineages (Fig. 2D). Our early (March–May 2020) sequences represented nine lineages, though the majority were classified as B.1 and B.1.1, the globally dominant lineages at the time[19].

Jordan initially contained community transmission after first detection in early March, with the weekly rolling average case numbers remaining below 25 until the middle of August 2020 (Fig. 1A). The high number of introductions without sampled onward transmission in our phylogeographic analyses supported the epidemiological data that showed a relatively high number of introductions contained by comprehensive mitigation strategies (Fig. 1C). Alongside travel restrictions and institutional quarantine, the epidemic was controlled with strictly enforced adherence to NPIs including a comprehensive curfew, strict gathering restrictions and school, retail and non-essential workplace closure implemented on 17 March 2020 (Fig. 1C).

In our phylogeographic reconstructions, we found no evidence that SARS-CoV-2 circulated cryptically in Jordan before identification of the first local cases. The first sampled introduction into Jordan was from Europe and occurred in mid-to-late February to early March across all datasets. This was consistent with epidemiological evidence that the first case returned from Italy (the epicenter of the European outbreak at the time) 2 weeks prior to detection on 2nd of March in quarantine (Supplementary Fig. 5). This suggests that Jordan had an efficient early surveillance program that initially contained the epidemic, though the role of chance cannot be discounted.

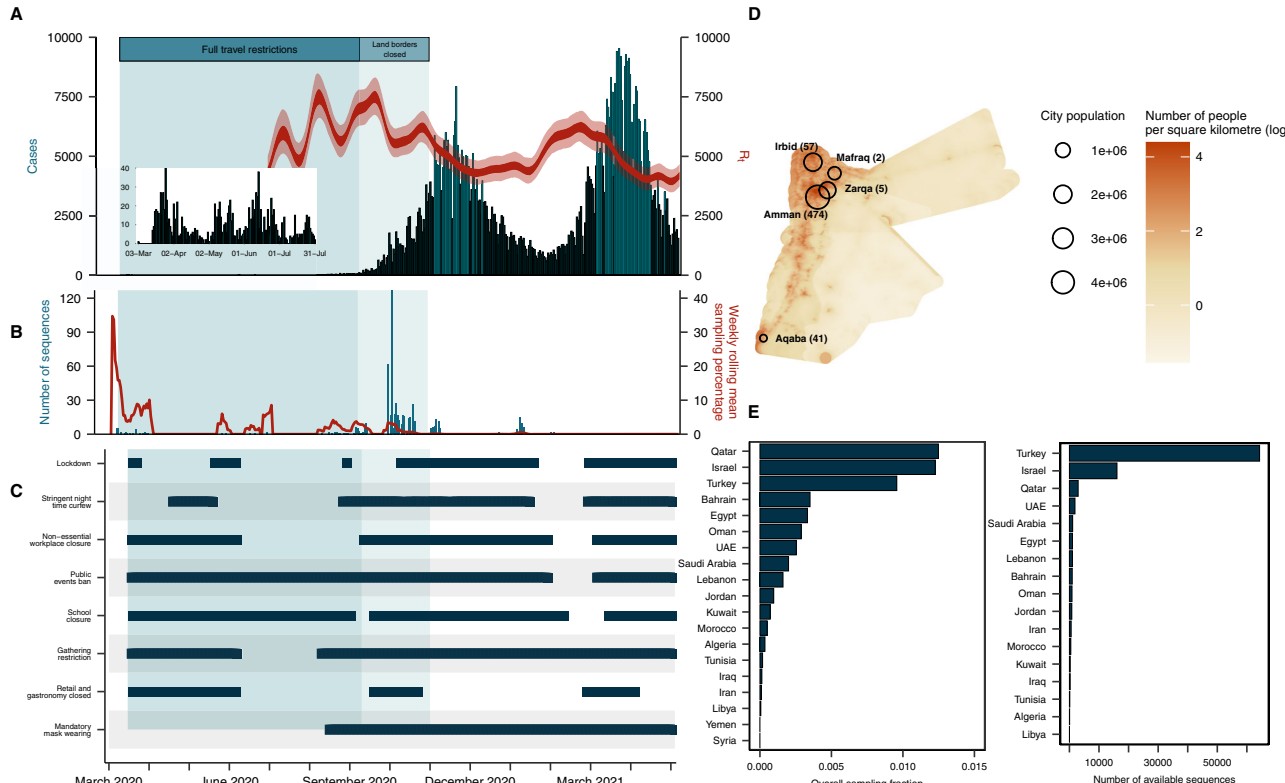

**Fig. 1 | Investigating the first year of the SARS-CoV-2 pandemic in Jordan.**
**A** Reported daily cases in Jordan (bar graph, left axis) and time-varying effective reproduction number (ribbon, right axis). **B** Number of sequences generated in this study from Jordan (bar graph, left axis) and weekly rolling mean sampling fraction for Jordan (line, right axis). **C** Timeline of non-pharmaceutical interventions in Jordan (see Methods for definitions). **D** Geographic distribution of sequences generated in the current study, relative to population density of Jordan and its major sampled cities. **E** Overall sampling fraction for all Middle Eastern countries and number of publicly available SARS-CoV-2 sequences on GISAID for Middle Eastern countries.

## The geographic origin of introductions into Jordan shifted to regional seeding during the period of travel restrictions

In our phylogeographic analyses of the major sources and timing of introductions into Jordan, we found that the Middle East was the second largest source of introductions (Fig. 2B, C). We found that the number of introductions from the Middle East increased relative to Europe across all datasets for the period travel restrictions were in place (mid-March to September 2020), suggesting a shift to regional seeding during this period (Fig. 3A). This trend was consistent across the downsampling strategies and randomizations (Supplementary Fig. 6).

To understand how undersampling in the Middle East affects the number of regional transmission events estimated, we developed three downsampling strategies, including two (III and RE, see Methods) that enriched for samples from the Middle East, and performed our phylogeographic analyses on three random replicates of each. The effect of downsampling on source-sink inferences was most pronounced for the Middle East estimates (Fig. 2B). We found that the estimated number of introductions from the Middle East were higher in III and RE datasets that had more representation from the region (Fig. 2B).

We estimated a low number of introductions during the travel restrictions period, which was expected given low incidence and therefore minimal sampling (Fig. 2C, D). Starting in August 2020, we observed the B.1.1.312 lineage dominating sampling, which fits with limited introductions due to travel restrictions and the onset of community transmission (Fig. 2D). In our phylogeographic analyses of B.1.1.312, we found that B.1.1.312 likely emerged in Jordan (see Supplementary Information for our full analyses). However, it is impossible to exclude neighboring Middle Eastern countries as the country of

origin due to regional sampling biases and the strong evidence of regional migration and connectivity supported in this work. Notably, B.1.1.312 has the Q957L substitution in a conserved heptad repeat region of the S2 subunit of the spike glycoprotein, which has been implicated in the conformational changes required for cell-cell fusion and viral entry for SARS-CoV-1 and HIV[20,21]. We characterized the phenotypic relevance of the Q957L substitution in silico and in vitro and found no functional advantage for Q957L, despite previous claims (see Supplementary Information)[22].

The number and origin of introductions estimated from genomic data is fundamentally dependent on the sample. To mitigate this sample-dependence, we investigated which countries are most likely to act as sources of transmission based on their connectivity to Jordan by quantifying changes in air and land travel to Jordan over time. We obtained travel data from the International Air Transport Association (IATA) and the Customs Agency of the Hashemite Kingdom of Jordan. All non-essential air travel to and from Jordan was banned on 17 March 2020. We found that the travel restrictions reduced air travel volume by 96% through April 2020 (Fig. 3B). We found that ~85% of incoming air passengers originated from countries in the Middle East from March to early September 2020, when travel restrictions were in place (Fig. 3B, C). This strongly supports regional connectivity as the dominant source of risk from air travel during this period, which is consistent with the introduction profile we estimated in our phylogeographic reconstructions (Fig. 3A).

We found that land travel volume declined by 99% starting in April 2020 for private vehicles and buses, predominantly originating in Saudi Arabia (Fig. 3D). Private land travel volume remained at 0.06–3.6% of pre-pandemic volume through December 2020 (Fig. 3D). Only select border crossings with Saudi Arabia, Israel and Palestine

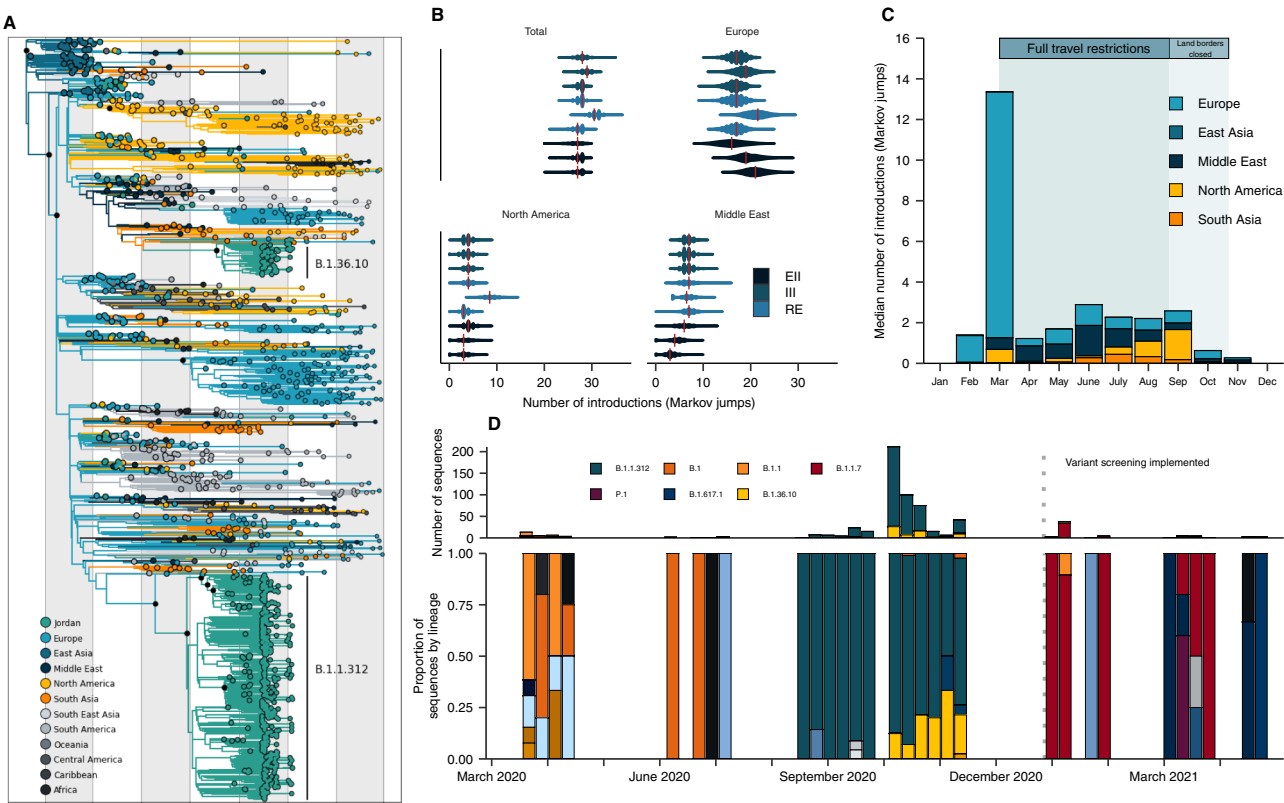

**Fig. 2 | Phylogeographic analysis of SARS-CoV-2 in Jordan. A** A representative time-calibrated phylogeny. Branches are colored by region-level geographic state reconstruction. Internal nodes annotated with black point represent posterior support >0.75. **B** The estimated number of introductions or Markov jumps into Jordan from source regions summarized across three replicates for the three downsampling strategies annotated in color. EII: Epidemiological incidence informed; RE: Regional enrichment; III: Introduction Intensity Index informed—see Methods for full definition. **C** Posterior median number of introductions into Jordan binned by month from each region, summarized across all datasets. **D** Lineage frequency profile over time for 579 sequences generated in this study, classified by the Pangolin nomenclature. Only major lineages in Jordan and variants of concern are listed in the legend. Dashed line indicates when variant screening was implemented.

opened from late October 2020, allowing 100–150 pre-registered travelers a day subject to testing and home quarantine[23]. This suggests that introduction risk from private land travel was limited for most of 2020. Alongside private land travel, Jordan has high levels of commercial truck (freight) volume in bidirectional transit with its neighboring countries. We found that incoming truck volume declined to 50–60% of the pre-pandemic baseline in April and May but recovered to largely pre-pandemic levels from July onwards (Fig. 3D). We found that Saudi Arabia had the highest number of incoming trucks, followed by Iraq and Israel-Palestine. The sustained levels of freight transport during the travel restrictions strongly suggest that a lot of the introduction risk in the region is mediated by freight trade.

### The geographic origin of introductions into Jordan shifted again after travel restrictions were lifted

In our phylogeographic analyses, we found that viral introductions from non-Middle Eastern countries increased again after travel restrictions and institutional quarantine were lifted on 8th and 23rd September 2020 respectively (Fig. 3A). The number of introductions from North America increased starting in September, though the absolute estimated number remained low. In the travel data, we found that air volume recovered to only 15–22% of pre-pandemic levels after travel restrictions were lifted in early September (Fig. 3B). Non-Middle Eastern air travel increased from 15 to 25% of total volume, predominantly driven by incoming travelers from the USA, which suggests an increased introduction risk from intercontinental air travel (Fig. 3B). Overall, we found little evidence of introduction risk from regions

other than Europe, North America and the Middle East across all phylogeographic reconstructions (Supplementary Table 2).

The introduction of the Alpha variant (B.1.1.7) initiated the larger second epidemic wave in Jordan, with cases surging from the end of January 2021 and peaking in mid-March 2021 at over 9500 daily cases (Fig. 1A). We identified Alpha variant sequences by S-gene target failure (SGTF) screening from December 2020 to January 2021 and performed an independent phylogeographic analysis of these sequences in a globally representative background sequence set (see Methods) (Supplementary Fig. 7). We estimated that the first introduction of Alpha into Jordan occurred in late November, with a minimum of 17 sampled introductions [posterior median, 95% HPD 12–22] in mid-to-late December 2021. All the introductions originated from Europe, supporting the shift in the origin of introduction dynamics after the travel restrictions were lifted (Fig. 3A). This is consistent given the high incidence of cases during the second wave in the UK at this time, which was driven by the local emergence of Alpha in September 2020. This is also consistent with the introduction of Alpha into several European countries within months of its first detection in the UK (Supplementary Fig. 7)[24].

### Regional connectivity drove import risk into Jordan during the period of travel restrictions

Introduction risk is not just driven by connectivity but is a product of the number of travelers and the number of cases in the connected countries likely to travel. To account for the size of the epidemics in understanding transmission risk from potential source countries, we

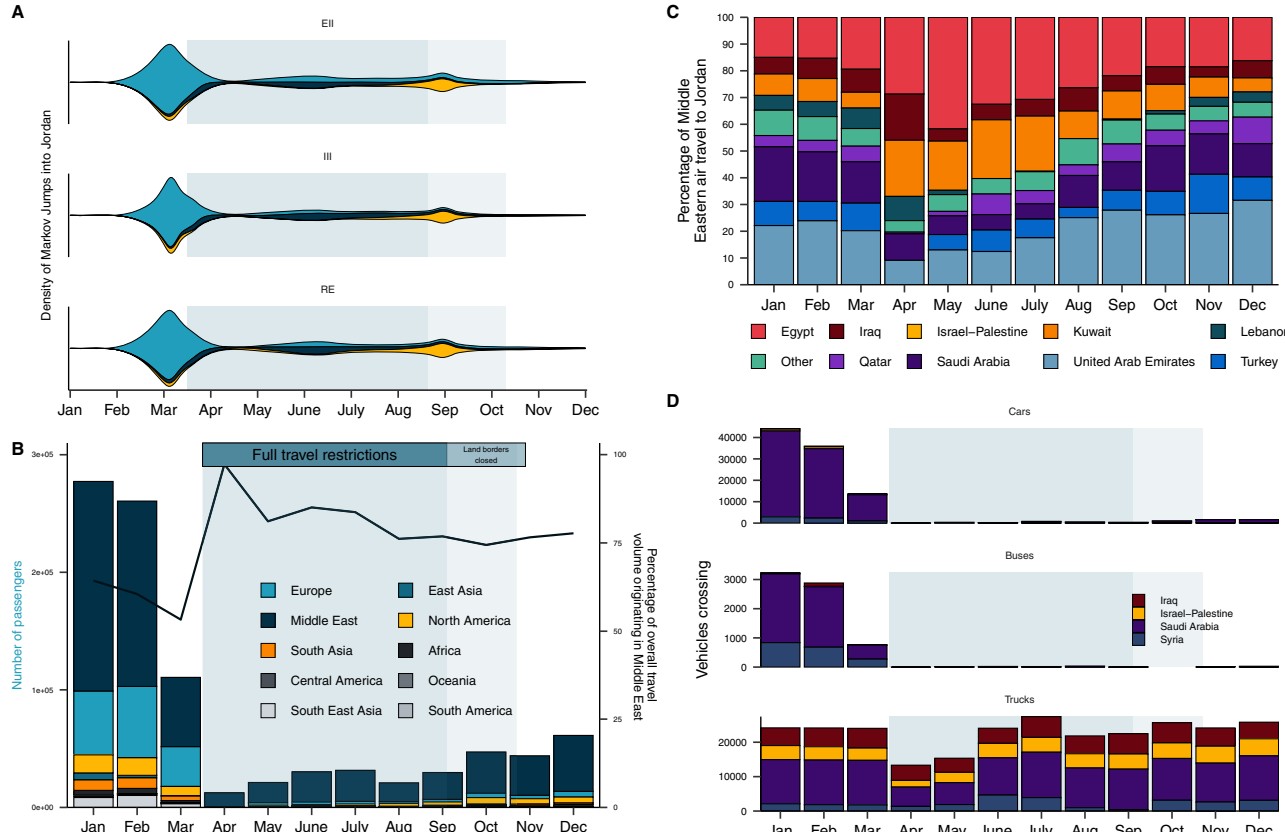

**Fig. 3 | Dynamic introduction risk shifts to regional seeding during the period of travel restrictions in 2020. A** The density of estimated introductions (Markov jumps) into Jordan over time from source regions by downsampling strategy (EII, III, RE) summarizing three replicates each. See Supplementary Fig. 6 for all replicates. EII: Epidemiological incidence informed; RE: Regional enrichment; III: Introduction Intensity Index informed. For legend see (**B**). **B** Volume of monthly air passengers destined for Jordan across 2020 by region (bar, left axis) and percentage of incoming volume originating from the Middle Eastern region (line, right axis). **C** Percentage of incoming air travel from the Middle East by country across 2020. **D** Incoming land-based travel by vehicle type for each bordering country across 2020. Time period of land border closure (light blue) annotated.

integrated the travel data with incidence data to quantify the importation (introduction) intensity index (II)[2]. The II quantifies the temporal trend in the daily estimated number of introductions into Jordan from source countries (see Methods).

We found that the dynamic II profile was highly consistent with the timing of the introductions from our phylogeographic reconstructions (Fig. 4A). The II peaked in early March 2020, driven by air travel from European countries experiencing high incidence[7]. We found that regional countries were not dominant source countries prior to the travel restrictions implemented on 17th of March despite high travel volumes as these countries largely had relatively lower incidence at the time (Supplementary Fig. 4)[25]. European countries' dominance in the II profile prior to the travel restrictions was consistent with the high number of introductions from Europe in March estimated in our phylogeographic reconstructions (Fig. 3A). We found that the total II expectedly reduced sharply in response to travel restrictions in March 2020, declining by ~65% from its early peak driven by Europe in March to the regional-driven peak during travel restrictions (17 March–8 September 2020) (Fig. 4A). The source risk profile shifted to Middle Eastern countries from late March till September, as these regional countries maintained higher volumes of both land and air travel into Jordan and experienced relatively delayed peak incidence (with some exceptions, including Israel and Iran)[25]. The temporal shift to regional risk in the II was consistent with the shift to higher relative regional seeding we estimated during the travel restrictions in the phylogeographic reconstructions (Fig. 3A). Saudi Arabia had the highest II across the entire period of travel restrictions, followed by Iraq and Israel-Palestine. To narrow down the source of

risk, we partitioned the II by travel and vehicle type (Fig. 4B). We found that the introduction risk from regional countries was predominantly driven by land travel, and in particular truck volume. We found that the II for non-regional countries, particularly the USA, increased after air travel restrictions were lifted in September 2020 (Fig. 4A). The increased II for the USA was consistent with the increase in introductions we observed from Northern America from September onwards in the genomic dataset. Consistent with the phylogeographic reconstructions, there was a negligible introduction risk into Jordan from regions other than Europe, North America and the Middle East.

As noted, the number of introductions estimated from genomic data is sample-dependent, with additional samples likely to reveal new introductions. To understand how sampling might result in misattribution of introductions in our genomic data, we aimed to formally compare the introduction profile estimated from genomic data to data streams that are not sample-dependent. We used dynamic time warping (DTW) to identify similarities in the temporal trends of introductions estimated from the genomic data and the estimated introduction index (II), which is sample-independent (see Methods). The DTW distance is minimized between time series with more similar trends[26].

We found that Europe's II trajectory had very similar temporal characteristics to the genomic estimates prior to the introduction of travel restrictions, consistent with our phylogeographic reconstructions (Figs. 3A and 4C). We found Europe's II trajectory had a substantially lower DTW distance to genomic estimates across all downsampling and randomizations compared to all other regions for the period prior to travel restrictions (Fig. 4D). This close synchrony

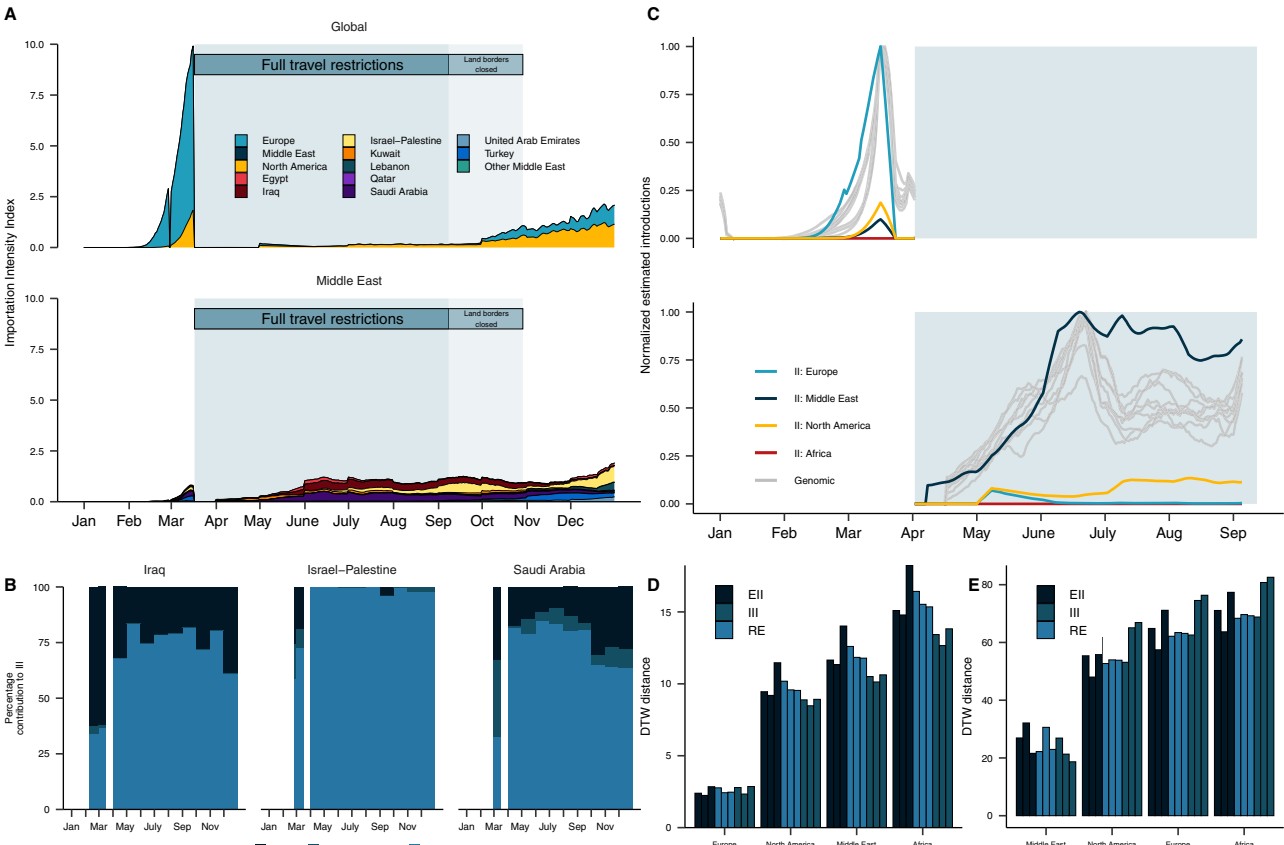

**Fig. 4 | Land-based regional connectivity drives introduction risk during the period of travel restrictions. A** Estimated introduction (importation) Intensity Index (II) for 2020, divided into the Middle East and the remaining (Global, non-Middle Eastern). Countries grouped by region, except Middle Eastern countries in the bottom panel, which are country-level. Legend in A applies to C. **B** Contribution of land and air travel to II in (**A**) for highest ranked regional countries. **C** Temporal synchrony between introductions estimated from genomic data (across all downsampling strategies and randomizations) and the weekly rolling average estimated introduction index (II) across the four regions summarizing the countries with the 20 highest individual-level II in (**A**) for the period prior to travel restrictions (top panel) and the period travel restrictions were in place (bottom panel). The normalized time series for the genomic estimates of introduction are depicted in light gray, with a line per seed and downsampling strategy, and the estimated introduction index (II) for each region are annotated in color. **D** The dynamic time warp distance between the query genomic estimate and each region's III across all downsampling strategies and randomizations for the period prior to travel restrictions. **E** The dynamic time warp distance between the query genomic estimate and each region's II across all downsampling strategies and randomizations for the period travel restrictions were in place.

between the trajectories—robust to all downsampling and randomizations—supports Europe as the primary source of introductions prior to travel restrictions. We observed highly similar temporal patterns between the genomic introductions estimates and the II trajectory of the Middle East for the period travel restrictions were in place (Fig. 4C, E). We found that the DTW distance of the Middle East's II to the genomic estimates was at least twofold lower than all other regions across all downsampling strategies and randomizations (Fig. 4E). This supports the shift to dominant regional seeding during the period the travel restrictions were in place, in line with our phylogeographic reconstructions. Taken together, the DTW analysis suggested good alignment between the source introduction profile estimated from the genomic data and the II, which consolidates travel data and the epidemiological curve of source countries. This suggested that our inferred profile from the genomic data is unlikely to be largely impacted by misattribution of sources owing to sampling biases, as the synchronous II is not subject to sampling biases.

### Border closures disproportionately reduced regional introduction risk

To determine the effect of the border closures on reducing transmission risk from imported cases in Jordan, we quantified the II under the assumption that travel volume was maintained at the levels of January–February 2020 for March–September (see Methods) (Fig. 5A).

We conclude that the Middle East, and in particular Saudi Arabia, had the largest reduction in introduction risk relative to our counterfactual. This was driven by the reduction of large volumes of private car and bus travel (Fig. 5C). This supports our findings that land travel is a major driver of introduction risk, and strongly suggests that introduction risk can be effectively reduced by interventions such as land border closures. Notably, introduction risk was not eliminated as commercial land travel was continued throughout this period (Fig. 4A, B). The reduction in non-Middle Eastern countries' aggregate II was driven almost exclusively by reduced introduction risk from the USA, as well as European countries during the early pandemic months. This supports previous evidence that there was negligible introduction risk to Jordan outside of Europe, North America, and the Middle East.

### Undersampling in the Middle East region obscures Jordan's role as a source of regional transmission

We aimed to better understand transmission in the Middle East by analyzing Jordan's role as a source of introductions to other Middle Eastern countries relative to the rest of the world. However, Jordan's export dynamics and the transmission dynamics of SARS-CoV-2 in the wider Middle East are obscured by unrepresentative, limited sampling. In our phylogeographic analyses of viral exports from Jordan, we only found evidence that Jordan acted as a source to Middle Eastern countries in the B.1.1.312 dataset, which is Jordan's most sampled

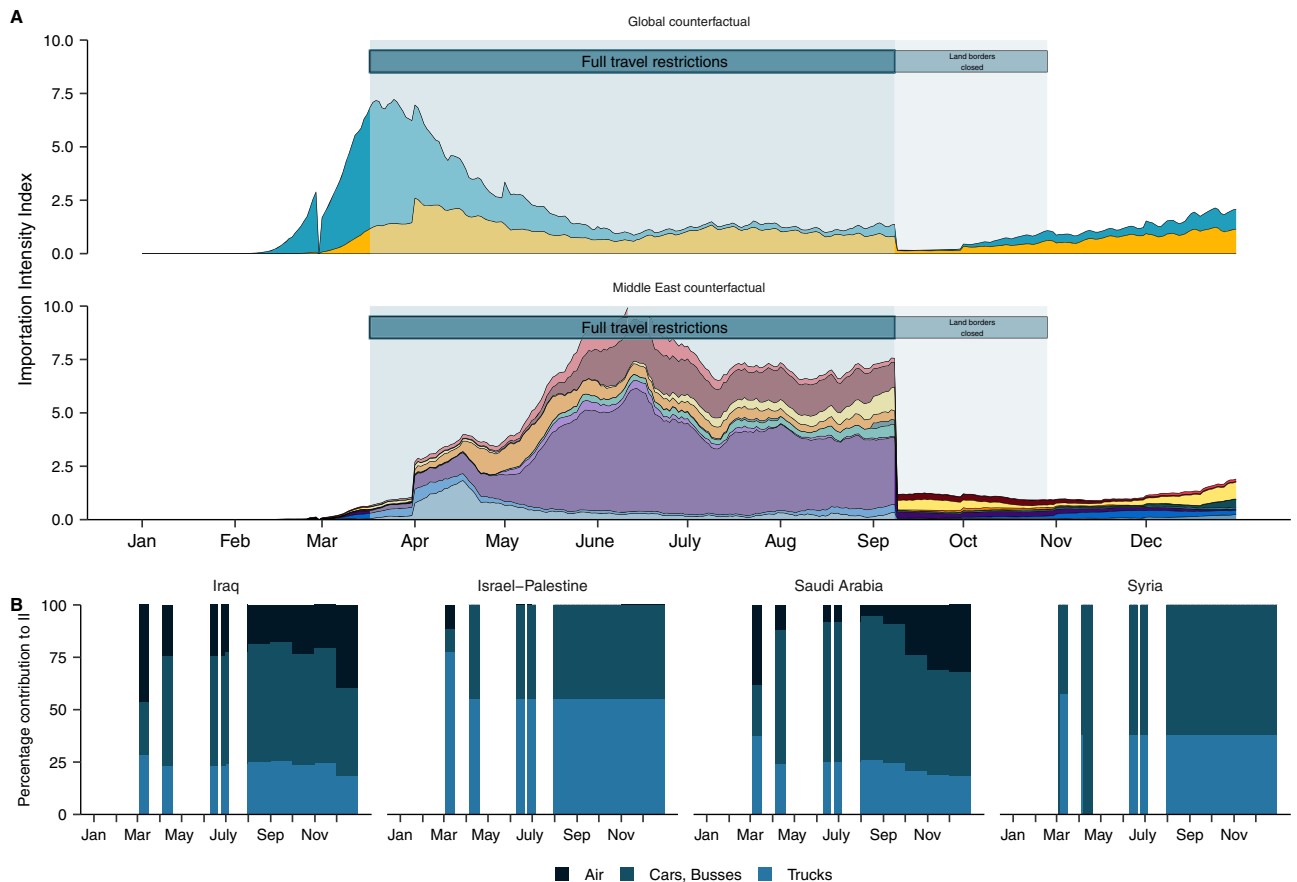

**Fig. 5 | Land-based regional connectivity drives introduction risk during the period of travel restrictions. A** The effect of travel restrictions on introduction risk. Counterfactual estimated Introduction Intensity Index (II) for 2020 should travel restrictions not have been implemented, assuming travel volumes of January–February were extended from March to September. The counterfactual II is annotated in faded colors. **B** Contribution of land and air travel to counterfactual II for highest ranked regional countries.

lineage owing to its predominance during its first wave. We only detected single exports of B.1.1.312 to Palestine, Egypt and Turkey respectively (See Supplementary information). We also did not find strong similarity in the lineage composition (see Methods) between Jordan and other Middle Eastern countries over time (Fig. 6A). However, it's highly unlikely that we would identify many transmission events as regional sampling fractions were particularly low at the time of Jordan's peak incidence (Supplementary Fig. 8).

To overcome these sampling biases, we quantified Jordan's exportation risk based on changes in outbound land and air travel for 2020. We estimated that 72% of air travel during the months of July–December 2020 (from the putative onset of local transmission– see Supplementary Information–to the end of the first wave) was bound for Middle Eastern countries, again highlighting the high level of regional connectivity (Fig. 6B, C). The USA was the only non-regional country in the top ten out-bound destinations. We found that out-bound private land travel decreased 99.9% toward April 2020, only recovering to 5% of the baseline by December 2020. The majority of land travel during Jordan's first wave flowed to Saudi Arabia and Syria (Fig. 6D). We found that the out-bound flow of freight transport declined to 51–58% of the pre-pandemic baseline in April–May 2020 but recovered to baseline by June 2020. The majority of freight transport was destined for Saudi Arabia (Fig. 6D). This was consistent with our analyses of introduction risk, with a large proportion of regional connectivity and therefore export risk driven by freight transport.

We investigated which countries were most likely to act as a sink for Jordan's exports across time by estimating Jordan's exportation

intensity index (EI). As with the II, this measure accounted for Jordan's incidence across time in combination with the number of outbound travelers to potential seed countries. We observed that regional connectivity had the largest contribution to Jordan's role as a potential source of transmission (Fig. 6E). We found that the EI peaked in November after onset in September, driven by Jordan's first wave (Fig. 1A). We found that exportation risk was far higher for Middle Eastern countries, collectively and individually, despite recommencement of air travel in early September. The EI was the highest for Saudi Arabia across the study period (Fig. 6E). Export risk for Saudi Arabia and all other neighboring countries was driven predominantly by high volumes of freight transport (Fig. 6F). The only non-regional country with a non-negligible EI was the USA. However, it is impossible to fully understand how Jordan's connectivity and export risk translates to establishing transmission chains of variable sizes in regional countries without more complete sampling (See Supplementary Information).

## Discussion

We integrated genomic, epidemiological and travel data to investigate the source-sink dynamics of SARS-CoV-2 in Jordan, focusing on Jordan's connectivity to the wider Middle East. Our results show how the source profile of introductions was dynamic over the pandemic. It shifted from a high number of introductions from Europe in the early pandemic months to lower levels of regional introductions for the travel restrictions period in both the phylogeographic reconstructions and the introduction intensity index. We also showed that land travel, and in particular freight transport, rather than air travel had the largest contribution to introduction risk during this period. Consistent with

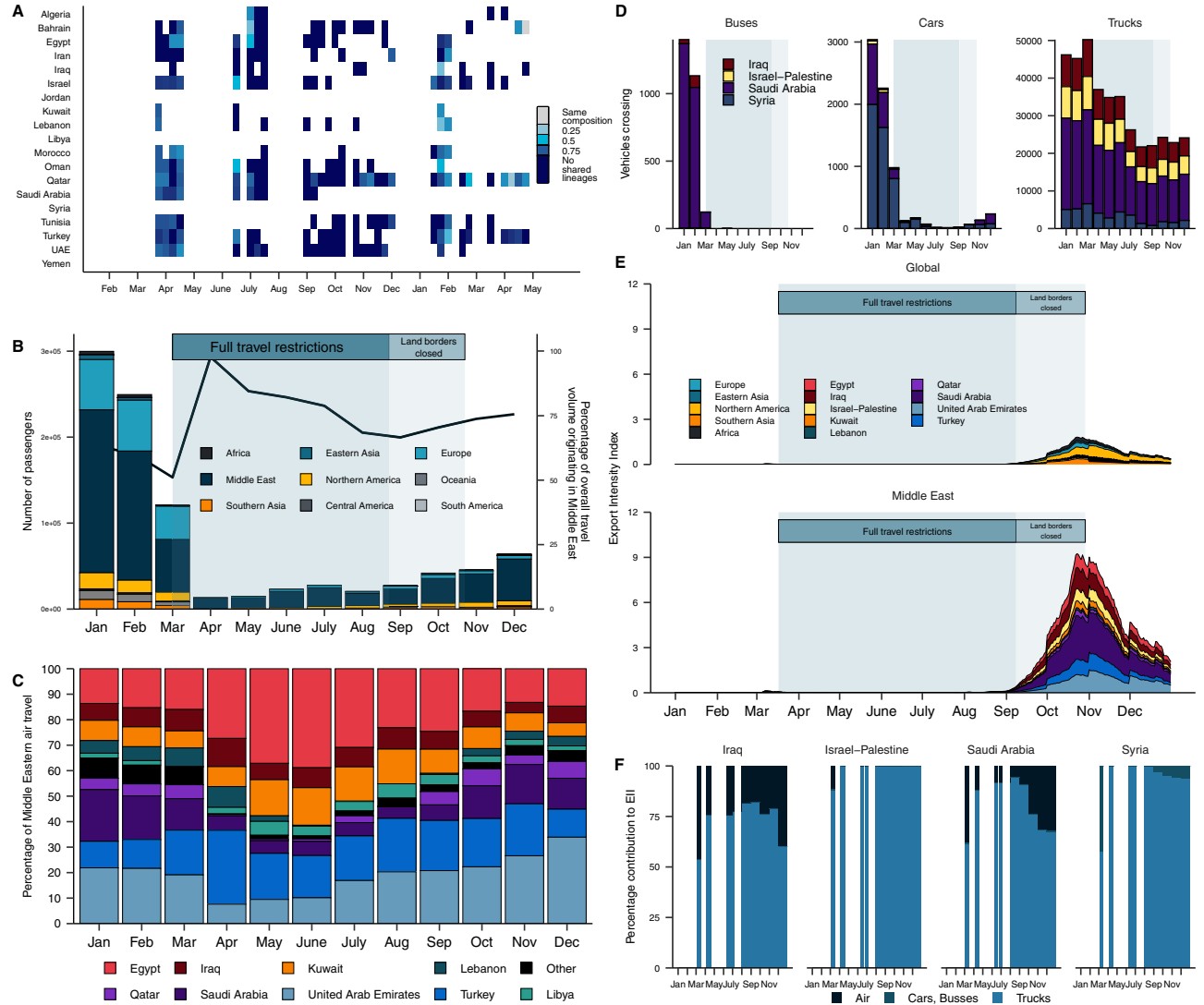

**Fig. 6 | Land-based regional connectivity drives export risk from Jordan.**
**A** Volume of monthly air passengers traveling from Jordan across 2020 by region (bar, left axis) and percentage of outgoing volume destined for the Middle Eastern region (line, right axis). **B** Percentage of outgoing air travel to the Middle East by country across 2020. **C** Outgoing land-based travel by vehicle type for each bordering country across 2020. Time period of land border closure (light blue) annotated. **D** Dissimilarity between weekly binned lineage frequency profiles as presence-absence of lineages between Jordan and its Middle Eastern neighbors by Bray–Curtis dissimilarity. Bray–Curtis dissimilarity is bound between 0 and 1, with 0 indicating the countries have the same composition and 1 indicates countries do not share any lineages. **E** Estimated Exportation Intensity Index for 2020. **F** Contribution of land and air travel to the EII of Jordan's neighboring countries.

the introduction risk profile, we showed that high regional connectivity disproportionately drove Jordan's export risk, with significant contribution from land-travel. Our findings for the Middle East are consistent with previous studies in Africa and Europe emphasizing regional connectedness and land travel as drivers of viral transmission[2,5,7,8].

Our findings emphasize that strategies aiming to stop or slow the spread of viral introductions, including new variants, with travel restrictions need to prioritize managing risk from land travel including commercial transport alongside air travel for restrictions to be effective[27]. Jordan's land crossings were only open for commercial cargo from March 2020 until late October 2020[28]. Jordan first mandated routine testing and institutional quarantine upon arrival at the border for all truck drivers on 15 April 2020 after infections were reported in truck drivers and their contacts[29,30]. However, early cases and superspreading events were continuously associated with truck drivers, particularly crossing from Saudi Arabia[30–35]. The early preventative measures were likely undermined by overcrowding at the border crossings, initial self-reported quarantining, and delays in

construction of the quarantine site at the Al-Omari border with Saudi Arabia, though unloading methods were adapted to minimize contact between drivers[36–38].

We did not identify many transmissions events from Jordan to other Middle Eastern countries, though our EI suggests continuous and relatively high export risk. However, it is impossible to know the full extent of transmission and the underestimation of transmission without more sampling in the region. This underestimation highlights how undersampling an entire region can obscure the importance of regional connectedness in sustaining local epidemics[5–7]. The introduction risk profile was robust to our downsampling strategies and randomizations. However, datasets with higher representation from Middle Eastern countries did have higher estimated introductions from the Middle East. This expected effect of downsampling strategy on source-sink dynamics emphasizes the need for caution in inference with a rapidly, globally dispersing pathogen subject to intense but biased sampling[1,2,6,39].

Our results should be interpreted in the context of our limited sampling, which represented 0.11% of all observed cases and was

disproportionate to epidemic incidence. Our introduction estimates are likely severely underestimated, given our small sample size and the negligible likelihood of observing singleton introductions or small-scale transmission chains[2]. The II quantifies the temporal trend in the daily estimated number of introductions rather than absolute values as it is based on the back-extrapolated time series of deaths and is therefore restricted by the associated reporting delays and biases. Notably, the II will be severely underestimated for countries such as Syria and Yemen with large-scale underascertainment[40]. Our counter-factual II for the travel restriction period assumed travel patterns for March to September are similar to January and February, which are potentially inflated by holiday travel patterns. It also does not consider restrictions applied by the origin of return-travel passengers, which would likely have disincentivized travel overall.

Surveillance infrastructure in Jordan and the wider Middle East is insufficient to characterize the transmission dynamics of SARS-CoV-2, including monitoring new variants of concern (VOC). We first detected the Omicron variant on 7 December 2021, with limited sequencing capacity to monitor its transmission dynamics. Considering the strong connectedness evidenced in this work, it is important that regional surveillance programs coordinate and pool resources to strengthen regional capacity. A recent study suggests that surveillance programs would need to sequence 300 representative cases to detect at least one sequence of a lineage with a frequency of 2% in the population with 95% probability[41]. This framework suggests that at least 0.5% of cases should be sequenced with rapid turnaround during periods of high incidence to detect new variants as they emerge. Current sampling efforts in Jordan as well as the wider Middle East fall far short of this goal.

As an international community, we need to invest in more equitably distributed, locally sustained surveillance infrastructure as well as diagnostic capacity that can scale in real-time, without the delays associated with dependence of low- and middle-income countries or Global South institutions on others. There are significant inequities in public financing, the social determinants of health and health system capacity across the Middle East, with public health efforts in several countries operating under conditions of conflict, displacement and extreme resource limitations[42,43]. Sustainably strengthening surveillance and diagnostic capacity across the Middle East and other regions worldwide is not only imperative to empower local public health decision making, but is also vital to monitoring the emergence of novel variants and zoonoses in a vulnerably connected world.

## Methods
### Ethical clearance
The study was approved by the Institutional Review Boards (IRBs) at Scripps Research Institute (TSRI) (IRB-21-7739) and the Cell Therapy Center of The University of Jordan (IRD-CTC/1- 2020/01). A consent waiver was requested and approved by IRB as this study uses pre-existing RNA obtained anonymously from various Biolab collection points, with no risk to subjects.

### Sample collection
We collected SARS-CoV2 samples from routine diagnostic tests performed by Biolab Diagnostic Laboratories, which has 20 branches in five governorates including major cities Amman, Irbid, and Zarqa. We collated samples from walk-in/drive-through testing centers, public and private referral laboratories, house-call services as well as health care facilities. We sequenced samples predominantly from the densely populated Amman (population > 4 million) and Irbid (population > 2 million), which were also the epicenters of the outbreak in Jordan across both waves in our study period. Five to ten percent of total samples with a Ct value <32 in the TaqPath COVID-19 PCR assay were selected at random for sequencing across the major cities. Screening for VOC by SGTF and mutation-specific primer sets was implemented

from late December 2020. Samples where all three target regions of the ORF-1, S, and N genes were amplified were deemed non-SGTFs, while those where ORF-1 and N gene target regions were amplified only, were classified as SGTFs and used subsequently for random sample selection for sequencing. In total, we generated 579 sequences before variant screening and 37 Alpha variant sequences during the period of this study. All sequence data generated were made available via the GISAID database (accessions available https://github.com/andersen-lab/HCoV-19-Genomics).

### SARS-CoV-2 whole-genome sequencing
SARS-CoV-2 was sequenced using PrimalSeq-Nextera XT. This protocol is based on the ARTIC PrimalSeq protocol and adapted for Illumina Nextera XT library preparation[44]. The ARTIC network nCoV-2019 V3 primer scheme uses two multiplexed primer pools to create over-lapping 400 bp amplicon fragments in two PCR reactions (available at https://github.com/artic-network/artic-ncov2019/blob/master/primer_schemes/nCoV-2019/V3/nCoV-2019.tsv). Instead of ligating Illumina adapters, Nextera XT is used to circumvent the 2 × 250 or 2 × 300 read length requirement. A detailed version of this protocol can be found here: https://andersen-lab.com/secrets/protocols/. Briefly, SARS-CoV-2 RNA (2 mL) was reverse transcribed with Super-Script IV VILO (ThermoFisher Scientific). The virus cDNA was amplified in two multiplexed PCR reactions (one reaction per ARTIC network primer pool) using Q5 DNA High-fidelity Polymerase (New England Biolabs). Following an AMPureXP bead (Beckman Coulter) purification of the combined PCR products, the amplicons were diluted and libraries were prepared using Nextera XT (Illumina) or NEBNext Ultra II DNA Library Prep Kits (New England Biolabs). The libraries were purified with AMPureXP beads and quantified using the Qubit High Sensitivity DNA assay kit (Invitrogen) and Tapestation D5000 tape (Agilent). The individual libraries were normalized and pooled in equimolar amounts at 2 nM. The 2 nM library pool was sequenced on an Illumina NextSeq using a 500/550 Mid Output Kit v2.5 (300 Cycles). A subset of samples from Ochsner Health were processed without tagmentation and sequenced on a Illumina MiSeq using a MiSeq reagent kit V3 (600 cycles). Raw reads were deposited under BioProject accession ID's PRJNA643575 and PRJNA612578. Consensus sequences were assembled using an inhouse Snakemake[45] pipeline with bwa-mem[46] and iVar v1.2.2[47].

### Downsampling strategies and dataset curation
Phylogeographic inferences are fundamentally dependent on and confounded by sampling biases and downsampling strategy. Models that explicitly incorporate travel history and unsampled lineages from undersampled populations are required to account for the rapid rate of viral dispersal and uneven sampling between locations that may result in unsampled intermediary locations[6,39]. However, we did not have access to the travel histories of our sequences. We therefore assessed the robustness of our findings over three random replicates of three downsampling strategies developed to minimize the impact of sampling biases in a computationally tractable dataset ($n = 1500$, $n = 1000$ background sequences, Supplementary Fig. 9).

The downsampling strategies are:
1. Epidemiological incidence informed (EII). Sequences were randomly subsampled proportional to country-specific incidence data binned by epidemiological week to maintain a realistic sampling time distribution and to attempt to make the number of sequences from each country proportional to the country's incidence for phylogeographic reconstruction. We obtained the time series of reported cases for all available countries using the outbreak.info R package (https://github.com/outbreak-info/R-outbreak-info), which aggregates epidemiological data from the COVID-19 data repository by the Center for Systems Science and Engineering at Johns Hopkins University[48,49].

2. The regional enrichment (RE) dataset corrected the subsampling probabilities of the EII dataset, increasing the sampling weights of countries from the Middle East or MENA region by a factor of five. The Middle East and North Africa region was defined as Algeria, Bahrain, Egypt, Iran, Iraq, Israel, Jordan, Kuwait, Lebanon, Libya, Morocco, Oman, Palestine, Qatar, Saudi Arabia, Syria, Tunisia, United Arab Emirates, Yemen and included Turkey.

3. The introduction Intensity Index informed (III) dataset. Sequences were randomly downsampled by country and epiweek with inclusion probabilities based on the estimated II (introduction intensity) (see section below).

The EII downsampling strategy is unbiased by hypotheses on transmission dynamics in the Middle East. The RE dataset is enriched for sequences from the Middle East, as Middle Eastern sequences are likely to be underrepresented in datasets randomly downsampled owing to the sparse sampling of the Middle East. The III dataset is informed by the quantified introduction risk to Jordan, maximizing the inclusion of sequences from source countries according to their relative risk of introduction to Jordan, assuming our travel data represents a highly accurate picture of in-bound traffic.

We selected sequences according to the downsampling strategies from all publicly available sequences in GISAID for the period up to 31 March 2021 after filtering out duplicates, sequences without complete sampling dates and sequences with >5% ambiguous nucleotides. Three random replicate datasets were conducted for each downsampling strategy. GISAID IDs and author acknowledgements are provided in Supplementary Table 4. In addition we reconstructed lineage-specific datasets for B.1.1.312 and B.1.36.10 (the two major circulating clades in Jordan's first wave) and B.1.1.7 (which drove Jordan's second wave) to ensure inclusion of all relevant sequences in more tractable datasets. We included all publicly available B.1.1.312 after filtering as above ($n = 474$) and B.1.36.10 ($n = 155$) sequences on GISAID in their respective datasets. For the B.1.1.7 dataset, we followed the EII downsampling approach set out above to generate three random replicates of $n = 500$.

## Phylogenetic analyses

We aligned the sequence datasets to the reference genome Wuhan-Hu-1 (GenBank: MN908947.3) using MAFFTv7.505[50] We masked the 5′ and 3′ UTRs as well as sites previously identified as potential sequencing errors or suspect homoplasies[51]. We reconstructed maximum likelihood phylogenetic trees for each dataset using IQTREE2 under ModelFinder[52,53]. We collapsed zero branch lengths to account for the large number of polytomies that characterize SARS-CoV-2 phylogenies. We explored temporal structure in our dataset across fixed (root MN908947.3) and optimized rooting strategies using temporal regression in TreeTime 0.8.3, excluding outlying sequences more than three interquartile ranges from the clock filter[54]. We assigned lineages to each sequence set with the Pangolin nomenclature tool (version 3.1.11)[18]. Lineage dissimilarity between countries was quantified as the bray–curtis distance between pairwise presence-absence matrices of lineage counts binned by epidemiological week using the *vegan* package in R[55].

We reconstructed time-scaled phylogenies for each dataset using BEAST 1.10.5[56]. We used the HKY substitution model with a gamma-distributed rate variation among sites across all of the global datasets[57]. We used a strict clock in the global datasets, with an informative lognormal prior and an exponential growth coalescent tree prior. We used an uncorrelated relaxed clock model with a lognormal prior in the smaller lineage-specific datasets. For each of the nine global datasets (non-lineage specific builds), we combined two independent MCMC chains of 200 million states ran with the BEAGLE computational library[56] Parameters and trees were sampled every 20,000 steps, with the first 20% of steps discarded as burn-in. For the lineage-specific datasets, we combined two independent chains of 100 million states, sampled every 10,000 steps with a burn-in of 10%. Convergence and mixing of the MCMC chains were assessed in Tracer v1.7, to ensure the effective sample size of all estimated parameters were >200[58].

We performed an asymmetric discrete trait analysis in BEAST version 1.10.5 with geographic states aggregated on a regional level to reconstruct the location-transition history across an empirical distribution of 4000 time-calibrated trees sampled from each of the posterior tree distributions estimated above[59]. Country-level traits were used for increased location history resolution for the more computationally tractable B.1.1.312 and B.1.36.10 lineage-specific datasets. We used Bayesian stochastic search variable selection to infer non-zero migration rates and identify the statistically supported transition routes into and out of Jordan by a Bayes Factor test[59].

Biases from unrepresentative sampling are compounded by uncertainty in the phylogenetic structure. SARS-CoV-2 phylogenies are often poorly supported by the posterior distribution owing to the low diversity in the sequence data, with poorly resolved regions (especially large polytomies) limiting robust reconstruction of evolutionary and phylogeographic relationships[1]. We included a Markov jump counting procedure to investigate the timing and origin of geographic transitions, or Markov jumps, into Jordan across the full posterior to account for uncertainty in phylogeographic reconstruction associated with sparse sampling and low sequence variability[17]. We used the Tree-MarkovJumpHistoryAnalyzer from the pre-release version of BEAST 1.10.5 to obtain the Markov jumps and their timings from posterior tree distributions[7]. We used TreeAnnotator 1.10 to construct Maximum clade credibility trees for all datasets. Trees were visualized using baltic (https://github.com/evogytis/baltic).

## Epidemiological data and estimation of effective reproduction number

We estimated the time-varying effective reproduction number with the Epinow2 package, fitting the time series of cases using Markov-chain Monte Carlo (MCMC) sampling, with 4 chains of 4000 samples and a warm-up of 1000. We obtained the time series of reported cases for Jordan using the outbreak.info R package[48] We assumed the same generation and reporting delayed distributions as[60] and assessed posterior convergence using the Rhat statistic. We only modeled the R(t) from the 3rd of July onwards—after the occurrence of the first 10 deaths—as this is the period when local transmission not introduction was assumed to be the dominant source of new infections[61].

## Data sources and timeline of interventions

We collated data on the timing and type of all major NPIs in Jordan for the entire study period. Data sources on the laws and measures executed included the official communications of the government of Jordan, including its Ministry of Health, as well as major media sources (See Supplementary Table 3 for the full timeline and reference sources). Major interventions of interest included comprehensive and stringent night time curfews, public event bans, gathering restrictions, school closures, workplace closures, mandatory mask wearing policies and the closure of specific economic sectors. The intervention timeline from primary sources was validated against the timeline from the Oxford Covid-19 Government Response Tracker (OxCGRT), with interventions conservatively considered to be in place if coded as "required" not "recommended"[62]. Conflicts were resolved using the primary sources.

## Travel and mobility data

We collated several independent data sources to understand Jordan's mobility and connectivity over time. We obtained air passenger volumes aggregated by month from the International Air Transportation Association (IATA) for January–December 2020. This dataset

included the number of incoming and outgoing passengers by origin/destination country to/from Jordan's two international airports, the Queen Alia International Airport in Amman and the King Hussein International Airport in Aqaba. Land-based travel volumes were obtained from the Customs Agency of the Hashemite Kingdom of Jordan via direct communication. This data included the number of private vehicles, buses and trucks entering and exiting Jordan aggregated by month. Jordan's border include crossings from Syria (Jaber, Ramtha crossing), Iraq (Al-Karamah Border Crossing), Israel and Palestine (Allenby/King Hussein Bridge, Sheikh Hussein crossing, Wadi Araba Crossing) and Saudi Arabia (Umari Border Crossing, Mudawara Border Crossing, Durra Border Crossing) and passenger ferries from Egypt. As there was no information on the complete route, vehicles were assumed to have originated only from the country of the crossing. For vehicles without an assigned origin, the nationality of the passengers were assigned if from the MENA region.

### Estimated introduction and exportation intensity index

We follow Du Plesis et al.[2] in calculating the introduction intensity index (II). Briefly, the II estimates the daily risk of introductions into Jordan from each source country as the product of the number of asymptomatically infectious individuals in each source country on that day (based on back-extrapolated death time series) and their likelihood to travel to Jordan based on the volume of in-bound land and air travel from the source country. We assumed the same estimates for the latent and incubation period, infectious duration, symptom-onset-to-death and asymptomatic proportion as[2] (See for full details). We obtained the time series of reported deaths from each country from the outbreak.info R package. To model incoming travel in the II, we combine the air and land-based volumes for each source country as applicable. Both the land and air travel data was aggregated by month. In the absence of additional information, we assumed that travel was uniform across all days in the month. The land-based travel dataset did not include the number of passengers in each car, bus or truck. We conservatively assume one passenger per truck, 1.5 passengers per car and 10 passengers per bus to quantify the number of inbound and outbound land-based passengers for the II. The II was quantified for January 2020 - December 2020. We did not consider traffic at Jordan's maritime border, which would underestimate Jordan's connectivity to Egypt. The export intensity index was qualified under the same assumptions for outgoing travel, integrated with Jordan's epidemiological curve. We examined the sensitivity of our II and EII results to the fixed parameters (asymptomatic proportion (ranging 0.2–0.5), generation time (4–10 days)). Results were robust over these ranges[2].

### Dynamic time warp analyses

Dynamic time warp (DTW) is a technique used to find optimal alignments between time-dependent data, acting as a measure of distance between time series accounting for noise, shifts and amplitude changes[26]. We performed DTW to quantify how similar the time series of introductions from the genomic data and the estimated introduction index (II) was. The DTW distance is minimized between time series with more similar trends. We used the normalized introductions over time estimated from the genomic dataset as the query series, categorized in the three phases under investigation: (1) prior to the implementation of travel restriction in March 2020, (2) while travel restrictions were in place from March 2020 to early September 2020 and (3) after travel restrictions were lifted. We used the *DTAI-Distance* module implemented for python[63] We have very few sampled introductions after travel restrictions were lifted in September, as sampling was dominated by the emergence of B.1.1.312 and onset of community transmission putatively in late June (see Supplementary Information). It is therefore not meaningful to calculate a distance for this period.

### Adaptive evolution of the Q957L substitution

We investigate the possibility that the Q957L substitution of the B.1.1.312 lineage resulted from diversifying positive selection using the MEME (Mixed Effects Model of Evolution) model in Hyphy v2.5.27[64,65].

### Structural analyses of the Q957L substitution

We used FoldX (http://foldxsuite.crg.eu/) to estimate the structural stability effects of Q957L. The spike protein structure (PDB: 7DWY, PDB DOI: 10.2210/pdb7DWY/pdb) was first repaired by removing potential steric clashes[66]. The difference in free energy changes arising from Q957L was then estimated under default parameters (298 K, ionic strength of 0.05 M and pH 7.0) using FoldX's empirical force field model which has a reported standard deviation of 0.46 kcal/mol between computed and experimentally measured values[67]. The mutation analysis was repeated five times and the mean and standard deviation free energy change values were reported.

### Experimental characterization of the Q957L substitution

**Cells and plasmids.** HEK293T (ATCC) and Calu-3 (ATCC) were cultured in Dulbecco's Modified Eagle Media (DMEM) supplemented with 10% Fetal Bovine Serum (FBS) and pen/strep/glut and maintained at 37 °C at 100% relative humidity.

The ancestral (WT) SARS-CoV-2 S cDNA was codon-optimized (GeneArt, ThermoFisher), synthesized as gene blocks (BioBasics), assembled by overlapping PCR, and cloned in pCAGGS. The Q957L mutation was introduced by overlapping PCR in the WT S or S bearing the D614G, also introduced by overlapping PCR. All S gene constructs were fully sequenced by Sanger sequencing.

**Lentiviral pseudotypes and transduction.** HEK293T cells were co-transfected with psPAX2 (gift from Didier 99 Trono, Addgene plasmid#12260), pLV-eGFP (gift from Pantelis Tsoulfas, Addgene plasmid # 36083) and plasmids encoding SARS CoV-2 S D614G, D614G and Q957L, or empty pCAGGS (mock) using JetPrime (Polyplus transfection, France) following the manufacturer's protocol. Twenty-four hours post-transfection, the medium was changed with DMEM 2% FBS pen/strep/glut. Media containing lentiviral particles were harvested at 48- and 72 h post-transfection. The viral producer cells were then washed with cold PBS and lysed (1%Triton X-100, 0.1% Igepal, 150 mM NaCl, 50 mM Tris-HCl, pH 7.5, 1X protease inhibitor cocktail (Cell Signaling)) on ice. Lysates were pre-cleared by centrifugation and proteins were denatured and resolved by SDS-PAGE. Proteins were then transferred onto PVDF membranes and S2 and GAPDH were detected by immunoblotting.

Media were filtered using 0.45 μm filters and directly used to transduce Calu-3 in the presence of 5 μg/mL polybrene. Forty-eight hours post-transduction, Calu-3 were trypsinized and resuspended in PBS-2% for flow cytometry analysis. The percentage of transduced cells was quantitated by measuring the GFP + cells.

**Cell-cell fusion assay.** The bi-molecular fluorescence complementation (BiFC) as been described previously*. Briefly, the HEK293T target cells were transfected using Jetprime with plasmids encoding the GCN4 leucine zipper-Venus1 (ZipV1, kind gift of Stephen W. Michnick, McGill University*), Ace2myc (pCEP4-myc-ACE2 was a gift from Erik Procko (Addgene plasmid # 141185)* and TMPRSS2 (from MISSION TRC3 Human LentiORF Puormicin Library, MilliporeSigma). The HEK293T effector cells were transfected using Jetprime with plasmid encoding GCN4 leucine zipper-Venus2 (ZipV2, also from Stephen W. Michnick) and pCAGGS encoding SARS-CoV-2 S WT, Q957L, or pCAGGS vector. Twenty-four hours post-transfection, cells were washed with PBS and detached with versene (PBS, 0.53 mM EDTA) and resuspended in serum- and phenol red-free DMEM. Wells of a 384-well black plates with optical clear bottom were seeded with effector and target cells (35,000 cells of each populations per well) and incubated

for 3 h at 37 °C, 5% $CO_2$. BiFC signal was acquired using Biotek Synergy Neo2 plate reader (BioTek Instruments) using monochromator set to excitation/emission of 500 and 542 nm.

## Reporting summary

Further information on research design is available in the Nature Research Reporting Summary linked to this article.

## Data availability

The data for our analyses can be found at https://github.com/andersen-lab/paper_2022_jordan-sars2-phylogenetics. All sequence data generated were made available via the GISAID database (accessions available https://github.com/andersen-lab/HCoV-19-Genomics) and NCBI under BioProject ID PRJNA612578. Travel data available on request owing to legal issues. Land travel data can be requested for release from customs@customs.gov.jo, with use pending approval by the Customs Agency of Jordan. Air travel data can be requested for release from Bluedot (info@bluedot.global), with use pending approval by Bluedot. Reference genome Wuhan-Hu-1 available at GenBank under accession MN908947.3. Oxford Covid-19 Government Response Tracker (OxCGRT) available at https://github.com/OxCGRT/covid-policy-tracker. Epidemiological information accessed from outbreak.info with https://github.com/outbreak-info/R-outbreak-info.

## Code availability

The code for our analyses can be found at https://github.com/andersen-lab/paper_2022_jordan-sars2-phylogenetics.

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

## Acknowledgements

We would like to acknowledge Mohammad Ghneim from Biolab's IT department for data extraction from LIMS, Fares Abu-Dayyeh from Jordan's Ministry of transport, Imad Ghuwairi from Jordan customs, and Daoud Sawaeer for facilitating the attainment of land travel data. We thank Diala Haddadin for collecting the raw data for the timeline of non-pharmaceutical public health interventions, Yuxia Bo and Justin Whitaker for technical help, and Vera A. Tang from the University of Ottawa Flow Cytometry and Virometry Core Facility for technical support. We thank the administrators of the GISAID database for supporting rapid and transparent sharing of genomic data during the COVID-19 pandemic and all our colleagues sharing data on GISAID. A full list acknowledging the authors submitting genome sequence data used in this study can be found in Supplementary Table 4. This work has been funded by CDC BAA contracts 75D30120C09795 (K.G.A.), NIH NIAID 3U19AI135995-03S2 (K.G.A.), U19AI135995 (K.G.A.), U01AI151812-02 (K.G.A.), NIH NCATS UL1TR002550 (K.G.A.), NIH NIAID R01 AI135992 (J.O.W.), NIH NIAI 1U01AI151378-01 (subaward Number 8-312-0217530-66439L) (I.A.D.). Experimental part conducted by Marceline Côté's laboratory was supported by a COVID-19 Rapid Research grant from the Canadian Institutes for Health Research (OV3 170632) to M.C. M.C. is a Canada Research Chair in Molecular Virology and Antiviral Therapeutics and a recipient of an Ontario Early Researcher Award.

## Author contributions

Conceptualization: E.P., M.Z., J.L.H., G.L., M.A.S., J.O.W., M.C., K.G.A., I.A.D. Methodology: E.P., C.A., G.L., K. G., M.A.S., J.O.W., M.C., K.G.A., I.A.D. Acquisition. A.T, R. R-S, A.Ariana, A.Awidi, S.A.J, E.K., M.C., A.Abdelnour, I.A.D. Data curation: E.P., C.A., M.Z., G.L., A.A.L., A.W., K.G., K.R., E.K., K.K., A.W., M.C., I.A.D. Analysis and interpretation: E.P., C.A., M.Z., A.T., J.L.H., G.L., M.B., K.G., E.K., N.L.M., A.X.H, M.A.S., J.O.W., S.W., M.C., K.G.A., I.A.D. Resources: A.Awidi, S.A.J., M.C., A.Abdelnour, K.G.A, I.A.D..Writing: E.P., M.Z., S.W., M.C., K.G.A., I.A.D. Review and editing: E.P., M.Z., J.L.H, K.G., M.A.S., J.O.W., S.W., M.C., K.G.A., I.A.D. Supervision: M.A.S., J.O.W., M.C., A.Abdelnour, K.G.A., I.A.D. Project administration: M.Z., C.A., R.R-S, L.D.H., M.M., E.S., L.N., M.C., I.A.D. Funding Acquisition: M.C., A.Abdelnour, K.G.A., I.A.D. All authors contributed to interpreting and reviewing the paper.

## Competing interests

The authors declare no competing interests.
