## [Peer Review File · Nature Communications]

Regional connectivity drove bidirectional transmission of SARS-CoV-2 in the Middle East during travel restrictionsREVIEWER COMMENTS

Reviewer #1 (Remarks to the Author):

This is a nice study that extends our knowledge on SARS-CoV-2 regional circulation in the Middle East during Mar 2020-Mar 2021 with Jordan as a main focus. A major strength of this manuscript is the use of multiple datasets integrating genomes with epidemiological datasets, including travel; use of multiple sampling techniques to compensate for poor regional sampling for Bayesian phylogenetics, and experimental methods to characterise mutations; to ultimately deliver a fine-scale characterisation of the timing, origins, mode of migration (via land, flight) under changing control measures and increasing restrictions on mobility. The authors find various improvements for the management of land travel, and identify gaps in surveillance and sequencing convincingly.

Overall, I think that the paper is well-presented and interesting and most of the limitations associated with the data are handled well. I only have minor comments.

1. Due to the predominance of some PANGO lineages over time, I wonder whether lines 93-97 can be improved to clarify the role of PANGO lineage designations to support introductions over time "We sampled a higher number of distinct lineages in the early pandemic months, in line with multiple independent introductions with limited onward transmission Figure 2D)"
2. In Line 137 "a strong sweep of diversity by the B.1.1.312 lineage" appears too strong a wording when there was little circulation or when low numbers were isolated previously.
3. I think, the conclusion that B.1.1.312 likely originated in Jordan (Line 139) can be expanded to allow neighbouring regions in the Middle East due to low sampling, and evidence of regional migration through border crossings.
3. Supporting information Line 10: suggest to formalise "phylogeographic build".
4. SI Line 26-27. Clarify whether line 26-27 relates to the R_t shown in Figures 1A. "We estimated an exponential growth rate of 12.1 [95% HPD 9.12 - 15.12] for B.1.1.312 in our phylodynamic models."

Reviewer #2 (Remarks to the Author):

In this manuscript, the authors describe the role of Jordan in the regional transmission dynamics of SARS-CoV-2, both in terms of introductions and exportations. The paper tries to address an important topic, namely how travel patterns and change of travel patterns impact regional transmission patterns. The paper is very well written and as such reads extremely well. I however have some major concerns at the moment with the results.

The first is that the metric for transmission between locations (number of introductions) is largely driven by sampling. The second one is that (at least the way I understand it), the arguments connecting Importation and Exportation risks to transmission dynamics are circular. In essence, the argument for why connectivity drove transmission is essentially that there is connectivity and therefore transmission must have also been driven by that. Or in other words, there isn't really a good connection between travel (II & EI) to transmission (phylogenetic analysis).

Major:

- I'm not convinced that the number of introductions is a useful metric to characterize importation/exportations dynamics in the first place. As shown in previous papers, local outbreaks in pretty much all locations originate from multiple introductions. Each introduction causes some sized outbreak and each additional sample has some probability of revealing a new introduction. There is therefore an extremely strong relationship between the number of samples and the number of introductions detected, meaning that the number of introductions is quite reflective of the number of samples. The same is true for the number of non-Jordan samples. Meaning, the more sequences from outside Jordan there are the more introductions into Jordan will be revealed and the more exportations will be revealed (as the authors also state). At the moment, I don't see

how any of this is accounted for and as such, I don't really see how 'number of introductions' tells us that much about the epidemic, other than what the number of samples was in different places. I would also suggest to add a description and discussion about whether the sampled sequences were in any way biased, i.e. was there preferential sampling of the air travelers or truck drivers and how this affected the inference results.

- The arguments around the Importation and Exportation risks and conclusions drawn from them seem a bit circular. The two indices are a function of incidence and travel data. If the travel changes or the incidence changes, naturally the indices would change too. If the goal is to learn how these changes in travel patterns actually impacted the spread of SARS-CoV-2, the metric to assess this, has to be the spread of SARS-CoV-2 itself and not the change of travel patterns. This is the case in particular for the last section of the results, where it's argued that Jordan should have a role as a source of regional transmission based on travel data even though there is no sequence data to support that. In light of that, I also think the title claiming that connectivity 'drove' transmission in the title (and elsewhere) should be adapted .
- As the number of introductions detected largely depends on the sampling, using II, EI or RE based subsampling will lead the estimations of introductions that confirm that these indices drive introductions. In other words, subsampling the dataset based on these indices, then using a highly sampling dependent metric to quantify transmission to confirm that these indices correlate with transmission is not the most convincing. Additionally, I would suggest to make the comparison between travel patterns and introduction patterns more statistical (by e.g. looking at correlation) instead of just qualitative.

I'm not sure how to address these best other than by i) showing that the estimated introductions and exportation patterns are not driven by sampling, or by correcting for those sampling biases in a way that doesn't just foreshadow the results (such as by using II weighted subsampling) and to ii) make an actual statistical comparison between travel patterns and introduction/exportation rates (not numbers).

Also, based on the lack of sampling in other places in the region, I think it is ok to not be able to say much about the exportation dynamics, but I would clearly state that the exportation dynamics are simply unknown because of a lack of data

Minor:

- 51: Rephrase to something like, due to a lack of sampling, we however didn't see any evidence for this.
- 86: number of introductions is somewhat meaningless without correcting for sampling, I would therefore not focus much on the numbers themselves.
- 102: Where do the 'rates' come from?
- 111: Why does it have to be due to the surveillance program, why not due to chance or other reason?
- 266: How are the sampling biases overcome by this?
- 282: There is no investigation to whether it 'drove' transmission
- 304: I would suggest to rephrase these sentences where it's stated that e.g. freight drove risk, as the indices are a function of those travel data.
- 311: There is no statistical analysis that truly connects travel patterns to transmission patterns, as such, I would be cautious to make PH recommendations.
- 324+: It's very likely that there are exportations from Jordan to neighboring countries, but as there is very little to actually show that I would suggest to write that it might/likely does obscure instead.
- 350: I'd rephrase that, as there is no modelling needed to compute that probability, which can be directly computed.
- 351: 0.5% seems like a bit arbitrary, why not more, or less

Reviewer #1 :

This is a nice study that extends our knowledge on SARS-CoV-2 regional circulation in the Middle East during Mar 2020-Mar 2021 with Jordan as a main focus. A major strength of this manuscript is the use of multiple datasets integrating genomes with epidemiological datasets, including travel; use of multiple sampling techniques to compensate for poor regional sampling for Bayesian phylogenetics, and experimental methods to characterise mutations; to ultimately deliver a fine-scale characterisation of the timing, origins, mode of migration (via land, flight) under changing control measures and increasing restrictions on mobility. The authors find various improvements for the management of land travel, and identify gaps in surveillance and sequencing convincingly.

Overall, I think that the paper is well-presented and interesting and most of the limitations associated with the data are handled well. I only have minor comments.

We thank the reviewer for their consideration and helpful comments.

Due to the predominance of some PANGO lineages over time, I wonder whether lines 93-97 can be improved to clarify the role of PANGO lineage designations to support introductions over time “We sampled a higher number of distinct lineages in the early pandemic months, in line with multiple independent introductions with limited onward transmission Figure 2D)”

We have clarified the statement in text in lines 103-105, which now reads: *“This supports the above evidence of multiple independent introductions of distinct lineages in the early stages with no one lineage dominating, indicating limited onward transmission or establishment of these lineages (Figure 2D).”*

In Line 137 “a strong sweep of diversity by the B.1.1.312 lineage” appears too strong a wording when there was little circulation or when low numbers were isolated previously.

We agree and have updated the text in line 155-157 to: *“Starting in August 2020, we observed the B.1.1.312 lineage dominating sampling”*

I think, the conclusion that B.1.1.312 likely originated in Jordan (Line 139) can be expanded to allow neighbouring regions in the Middle East due to low sampling, and evidence of regional migration through border crossings.

We agree and have extended the text in lines 159-161: *“However, it is impossible to exclude neighboring Middle Eastern countries as the country of origin due to regional sampling biases and the strong evidence of regional migration and connectivity supported in this work.”*

We also added text in the Supplementary Information section on B.1.1.312 to reflect this, in reference to the newly added Figure 7, in lines S28-34: *“Notably, several Middle Eastern countries were experiencing their first wave’s exponential growth (Iraq, Israel), peak (Egypt, Saudi Arabia), initial declines from their peak (Qatar, UAE) or prolonged waves during the emergence of B.1.1.312 (Kuwait) (Figure 7). These high case numbers in neighboring countries at the time of B.1.1.312’s putative emergence in Jordan, together with the regional sampling biases and the strong evidence of regional migration and connectivity supported in*

this work makes it impossible to exclude neighboring Middle Eastern countries as the country of origin.”

Supporting information Line 10: suggest to formalise “phylogeographic build”.

We have updated the text to “*lineage-specific phylogeny*” in line S14.

4. SI Line 26-27. Clarify whether line 26-27 relates to the R_t shown in Figures 1A. “We estimated an exponential growth rate of 12.1 [95% HPD 9.12 - 15.12] for B.1.1.312 in our phylodynamic models.”

We have updated the text in line S 38:39 to “We estimated an exponential growth rate of 12.1 [95% HPD 9.12 - 15.12] for B.1.1.312, **parametrized** in our phylodynamic models.”

Reviewer #2:

In this manuscript, the authors describe the role of Jordan in the regional transmission dynamics of SARS-CoV-2, both in terms of introductions and exportations. The paper tries to address an important topic, namely how travel patterns and change of travel patterns impact regional transmission patterns. The paper is very well written and as such reads extremely well. I however have some major concerns at the moment with the results.

We thank the reviewer for their consideration and helpful suggestions for improvements.

The first is that the metric for transmission between locations (number of introductions) is largely driven by sampling. The second one is that (at least the way I understand it), the arguments connecting Importation and Exportation risks to transmission dynamics are circular. In essence, the argument for why connectivity drove transmission is essentially that there is connectivity and therefore transmission must have also been driven by that. Or in other words, there isn't really a good connection between travel (II & EI) to transmission (phylogenetic analysis).

I'm not convinced that the number of introductions is a useful metric to characterize importation/exportations dynamics in the first place. As shown in previous papers, local outbreaks in pretty much all locations originate from multiple introductions. Each introduction causes some sized outbreak and each additional sample has some probability of revealing a new introduction. There is therefore an extremely strong relationship between the number of samples and the number of introductions detected, meaning that the number of introductions is quite reflective of the number of samples. The same is true for the number of non-Jordan samples. Meaning, the more sequences from outside Jordan there are the more introductions into Jordan will be revealed and the more exportations will be revealed (as the authors also state). At the moment, I don't see how any of this is accounted for and as such, I don't really see how `number of introductions` tells us that much about the epidemic, other than what the number of samples was in different places.

We appreciate the concerns of the reviewer, which we have addressed in more detail below.

We agree this is a foundational limitation of estimates relying on genomic data when sampling is restricted, as emphasized in line 168-169: *"The number and origin of introductions estimated from genomic data is fundamentally dependent on the sample."* To address this, in our previous manuscript we supplemented estimates of introductions based on the genomic data with analyses of travel data but more importantly, the estimated introduction index (II) and estimated export index (EI). These estimates do not rely on genomic samples as they are a product of travel data (for which we have near-complete data from the IATA and Jordan's customs agency, line 609+) and incidence, modeled to, among other things, account for the serial interval. The genomic estimates and non-sample dependent estimates from travel and incidence show high concordance on the pattern of the number and source of introductions into Jordan. We discussed the strong consistency between the sample-dependent genomic estimates and sampling-independent import/export estimates analyses from line 241 onwards.

In response to the reviewers' comments, we revised our manuscript to include a formal comparison of the introduction profile for the different source regions over time estimated from genomic data to data streams that are not sample dependent, alongside previous sample-independent estimates from the introduction index (II). We did an additional dynamic time warping (DTW) analysis (lines 267-292, together with updated Figure 4) to show strong synchrony in the profile of introductions into Jordan between the genomic data and the sample-independent estimated introduction index (II).

We believe the temporal patterns between the genomic estimates and the II for each source country/region is the most informative comparison. In contrast, assessing the temporal alignment between the travel data alone and the genomic introduction estimates is less informative than the II as it does not account for epidemic size (i.e. likelihood of travelers to be infected) in the source country. For example, we previously found that Middle Eastern countries had high, sustained levels of travel to Jordan prior to travel restrictions (**Figure 1 A below**). However, we found that these countries largely had far lower case numbers than North America and Europe at the time (**Figure 2 A below**) and were unlikely to have had a substantial contribution to introduction risk as quantified in the II as the countries with the higher case numbers at the time (Iran, Turkey) did not have the highest travel volume to Jordan (**Figure 1B - also see Main text figure 3B-D**). This motivates our inclusion of the DTW analyses between the II and the genomic introduction estimates rather than travel data in the revised main text. During the period of travel restrictions, we found that the Middle East region was the only region with sustained travel, resulting in the closest alignment between the Middle East travel time series and the estimated introductions (**Figure 1 C, D**). This recapitulates the result using the II (**Figure 4** in the main text), which is expected as the II incorporates travel information.

Figure 1: Temporal patterns between introductions estimated from genomic data (across all downsampling strategies and randomizations – 9 datasets) and the travel data across the four regions summarizing the countries with the 20 highest individual-level EII (see Figure xx) for Phase 1 (**A-B**), Phase 2 (**C-D**) and Phase 3 (**E-F**). **A, C, E** shows the normalized time series for the genomic estimates of introduction (in light gray) and the travel for each region. **B, D, F** shows the associated dynamic time warp distance between the query genomic estimate and each region’s travel data across all downsampling strategies and randomizations. The analyses after travel restrictions were lifted is limited by the low number of introductions estimated from the genomic data.

Figure 2: A) Weekly rolling average across regions before travel restrictions were implemented in Jordan. B. Normalized case numbers across the Middle East by country.

We have included the new DTW analyses in our revised main text from lines 267-292 and Figure 4. As noted in the text (lines 288 to 292): *“Taken together, the DTW analysis suggested good alignment between the source introduction profile estimated from the genomic data and the II, which consolidates travel data and the epidemiological curve of source countries. This suggested that our inferred profile from the genomic data is unlikely to be largely impacted by misattribution of sources owing to sampling biases, as the synchronous II is not subject to sampling biases.”*

NEW Figure 4: Land-based regional connectivity drives introduction risk during the period of travel restrictions. **A)** Estimated introduction (importation) Intensity Index (II) for 2020, divided into the Middle East and the remaining (Global, non-Middle Eastern). Countries grouped by region, except Middle Eastern countries in the bottom panel, which are country-level. Legend in A applies to C. **B)** Contribution of land and air travel to II in A for highest ranked regional countries. **C)** Temporal synchrony between introductions estimated from genomic data (across all downsampling strategies and randomizations) and the weekly rolling average estimated introduction index (II) across the four regions summarizing the countries with the 20 highest individual-level II in A for the period prior to travel restrictions (top panel) and the period travel restrictions were in place (bottom panel). The normalized time series for the genomic estimates of introduction are depicted in light gray, with a line per seed and downsampling strategy, and the estimated introduction index (II) for each region are annotated in color. **D)** The dynamic time warp distance between the query genomic estimate and each region's EII across all downsampling strategies and randomizations for the period prior to travel restrictions. **E)** The dynamic time warp distance between the query genomic estimate and each region's EII across all downsampling strategies and randomizations for the period travel restrictions were in place.

I would also suggest to add a description and discussion about whether the sampled sequences were in any way biased, i.e. was there preferential sampling of the air travelers or truck drivers and how this affected the inference results.

This is an important point and we apologize for not making this clear in the previous version of our manuscript. We have updated lines 61-64 accordingly to emphasize that we used random sampling from routine diagnostic testing without any bias towards travelers: “Towards this, we generated 579 sequences from 16 March 2020 to 31 December 2020 (encompassing the epidemic containment phase and the first epidemic wave) randomly sampled from routine diagnostic tests performed by Biolab Diagnostic Laboratories in four governorates”. This is now also reflected in lines 459-461 in the methods section: “We collected SARS-CoV2 samples from routine diagnostic tests performed by Biolab Diagnostic Laboratories, which has 20 branches in five governorates including major cities Amman, Irbid, and Zarqa. We collated samples from walk-in/drive-through testing centers, public and private referral laboratories, house-call services as well as health care facilities. “

The arguments around the Importation and Exportation risks and conclusions drawn from them seem a bit circular. The two indices are a function of incidence and travel data. If the travel changes or the incidence changes, naturally the indices would change too. If the goal is to learn how these changes in travel patterns actually impacted the spread of SARS-CoV-2, the metric to assess this, has to be the spread of SARS-CoV-2 itself and not the change of travel patterns.

The goal of the indices was to estimate the number of introductions/exports using data (travel, incidence data) that was not subject to the foundational sampling biases described by the reviewer. In the revised manuscript, we formalized the previously descriptive comparison between independent data streams using the DTW analyses described above. We do not believe the argument is circular at all, as the high agreement between the sample-independent indices and the estimates from the genomic data support our argument. Additionally, the indices estimate introductions per day, and are not intended to reflect the extent to which each introduction contributed to the local epidemic spread. This would be impossible to estimate without more complete epidemiological information and sequencing.

This is the case in particular for the last section of the results, where it's argued that Jordan should have a role as a source of regional transmission based on travel data even though there is no sequence data to support that.

In the revised version of our manuscript, we have emphasized this important point in line 334-336: “However, Jordan's export dynamics and the transmission dynamics of SARS-CoV-2 in

the wider Middle East are obscured by unrepresentative, limited sampling.” and line 409-411 “However, it is impossible to know the full extent of transmission and the underestimation of transmission without more sampling in the region.”

To address this further, in the revised version of the manuscripts, we analyzed the synchrony of the epidemiological curves of Middle Eastern Countries to Jordan’s epidemiological curve and estimated export risk (measured by the EI) to investigate whether Jordan’s estimated exports disproportionately impacted the spread of SARS-CoV-2 in the regional countries. This analysis can be found in a Supplementary Information Section (line S117+) and new Supplemental information figure 3. The new descriptive analysis shows that it is unlikely that Jordan’s export risk resulted in a B.1.1.312-driven wave in regional countries: *“suggesting Jordan’s export risk (Figure 6E) did not translate to establish dominant community transmission in neighboring countries. However, without more complete sampling, it is impossible to know the full extent of successful viral export.”* (line 404-408) We have stated in the Main text line 370-372: *“However, it is impossible to fully understand how Jordan’s connectivity and export risk translates to establishing transmission chains of variable sizes in regional countries without more complete sampling (See **Supplementary Information**).”*

In light of that, I also think the title claiming that connectivity ‘drove’ transmission in the title (and elsewhere) should be adapted .

We hope the reviewer can agree that our updated results prove that regional connectivity drove introductions and exports during the period of travel restrictions. Should the reviewer prefer, however, an alternative title would be *“Regional connectivity drove **risk** of bidirectional transmission of SARS-CoV-2 in the Middle East during travel restrictions”*.

As the number of introductions detected largely depends on the sampling, using II, EI or RE based subsampling will lead the estimations of introductions that confirm that these indices drive introductions. In other words, subsampling the dataset based on these indices, then using a highly sampling dependent metric to quantify transmission to confirm that these indices correlate with transmission is not the most convincing.

As other’s previous work^{1,2} has shown, models that explicitly incorporate travel history and unsampled lineages from undersampled populations are required to account for the rapid rate of viral dispersal and uneven sampling between locations that may result in unsampled intermediary locations. However, we did not have access to the travel histories of our sequences, which is why we designed three different downsampling strategies.

As described in the previous manuscript (line 493+): The RE dataset is enriched for sequences from the Middle East, which may be underrepresented in datasets randomly downsampled owing to the sparse sampling of the Middle East. This strategy was attempted to maximize inclusion of regional sequences as we were particularly interested in the regional contribution to the introduction/export dynamics. The Introduction Intensity Index (II) dataset maximizes the inclusion of sequences from source countries according to their relative risk of introduction to Jordan. This dataset was attempted to maximize the inclusion of sequences from countries with an evidenced risk of introduction to Jordan, supported by travel data and their epidemic situation over time. Most importantly, we also included a dataset set with sequences randomly subsampled proportional to country-specific incidence data binned by epidemiological week to maintain a realistic sampling time distribution.(Lemey et al. 2021) This dataset is agnostic to any of the indices mentioned above, and aimed to make the number of sequences from each country proportional to the country’s incidence.

We believe we address the sampling biases described throughout as robustly as possibly by assessing the robustness of our findings over three random replicates of all three downsampling strategies developed (line 499-501), including three datasets of the agnostic downsampling that are not subject to the circular argument. The results are highly similar across all three datasets, as illustrated in Figure 2B, Figure 3A and Supplementary figure 3 and discussed from lines 146-152 (new line numbers) and 413-417 in the previous manuscript.

We have clarified the importance of the different datasets in lines 522-528: *“The EII downsampling strategy is unbiased by hypotheses on transmission dynamics in the Middle East. The RE dataset is enriched for sequences from the Middle East, as Middle Eastern sequences are likely to be underrepresented in datasets randomly downsampled owing to the sparse sampling of the Middle East. The III dataset is informed by the quantified introduction risk to Jordan, maximizing the inclusion of sequences from source countries according to their relative risk of introduction to Jordan, assuming our travel data represents a highly accurate picture of in-bound traffic.”*

Additionally, I would suggest to make the comparison between travel patterns and introduction patterns more statistical (by e.g. looking at correlation) instead of just qualitative.

We agree and have included the dynamic time warping (DTW) analyses described above.

I'm not sure how to address these best other than by i) showing that the estimated introductions and exportation patterns are not driven by sampling, or by correcting for those sampling biases in a way that doesn't just foreshadow the results (such as by using II weighted subsampling) and to ii) make an actual statistical comparison between travel patterns and introduction/exportation rates (not numbers).

As described above, we believe the sample-independent introduction and export indices (II/EI) estimate introductions and exportation patterns in a manner not driven by sampling, as the indices are estimated from travel and epidemic incidence data. As described above, all our estimates are robust across three replicates of three downsampling strategies, notably including three replicates of the sampling strategy that is purely proportional to global epidemic incidence and agnostic to e.g. II weighted subsampling.

In response to the reviewer, in our revised manuscript we have now shown high agreement with a formalized DTW comparison between the genomic and II indices using the dynamic time warp analyses. Taken together, the consistency of our results across multiple lines of evidence (the travel data, the associated estimated introduction index and the genomic estimates) in high agreement (formalized with the dynamic time warp analyses) and robust to downsampling (across three replicates of informed and agnostic strategies) gives us confidence in our reported results.

Also, based on the lack of sampling in other places in the region, I think it is ok to not be able to say much about the exportation dynamics, but I would clearly state that the exportation dynamics are simply unknown because of a lack of data.

As previously mentioned, we have emphasized these important points in lines 334-336: *“However, Jordan's export dynamics and the transmission dynamics of SARS-CoV-2 in the*

wider Middle East are obscured by unrepresentative, limited sampling.” and line 409-411 “However, it is impossible to know the full extent of transmission and the underestimation of transmission without more sampling in the region.” We have also additionally analyzed whether Jordan’s export risk was realized in a B.1.1.312-driven wave in neighboring countries as described above (Supplementary Information lines 117+).

51: Rephrase to something like, due to a lack of sampling, we however didn’t see any evidence for this.

We have rephrased line 52-53 to “This was not evident in the genomic data due to a lack of sampling.”

102: Where do the ‘rates’ come from?

We have updated the text to clarify the colloquial use of “rates” to “number”.

111: Why does it have to be due to the surveillance program, why not due to chance or other reason?

We agree and have expanded line 123-125 to account for this: “This suggests that Jordan had an efficient early surveillance program that initially contained the epidemic, though the role of chance cannot be discounted.”

266: How are the sampling biases overcome by this?

The travel data is not subject to the same sampling biases as the sparse genomic data, especially as we had near-complete travel data from IATA and Jordan’s customs agency (lines 610+).

282: There is no investigation to whether it ‘drove’ transmission

We have updated lines 363-364 to “ We observed that regional connectivity had the largest contribution to Jordan’s role as a potential source of transmission”

304: I would suggest to rephrase these sentences where it’s stated that e.g. freight drove risk, as the indices are a function of those travel data.

We have updated lines 389-390 to “We also showed that land travel, and in particular freight transport, rather than air travel had the largest contribution to introduction risk during this period. “

311: There is no statistical analysis that truly connects travel patterns to transmission patterns, as such, I would be cautious to make PH recommendations.

We agree with the reviewer and have removed the line: “*Future and current containment strategies that rely on travel restrictions may benefit from targeting truck drivers for free rapid antigen testing programs.*”

324+: *It’s very likely that there are exportations from Jordan to neighboring countries, but as there is very little to actually show that I would suggest to write that it might/likely does obscure instead.*

We agree with the reviewer and have extended the text at lines 409-411 to “*However, it is impossible to know the full extent of transmission and the underestimation of transmission without more sampling in the region.*”

350: *I’d rephrase that, as there is no modelling needed to compute that probability, which can be directly computed.*

We agree and have updated the text to “*A recent study*” in line 437.

351: *0.5% seems like a bit arbitrary, why not more, or less*

This is in reference to the cited study, not our own suggestion as indicated in line 439 “*This framework suggests*”.

1. Lemey, P. *et al.* Accommodating individual travel history and unsampled diversity in Bayesian phylogeographic inference of SARS-CoV-2. *Nat. Commun.* **11**, 5110 (2020).
2. Butera, Y. *et al.* Genomic sequencing of SARS-CoV-2 in Rwanda reveals the importance of incoming travelers on lineage diversity. *Nat. Commun.* **12**, 1–11 (2021).

REVIEWERS' COMMENTS

Reviewer #2 (Remarks to the Author):

The authors addressed all my concerns, and I have only one suggested addition left regarding sampling.

* I may have missed that one, but it would be great to see how the sampling numbers for different locations over time differ for the different subsampling strategies (i.e. to show that they substantially differ).

Minor:

- 280: After this period?
- Figure 4 label: Should it be II instead of the EII?
- DTW: is the DTW affected by different absolute values for the II?

Reviewer #2:

The authors addressed all my concerns, and I have only one suggested addition left regarding sampling.

* I may have missed that one, but it would be great to see how the sampling numbers for different locations over time differ for the different subsampling strategies (i.e. to show that they substantially differ).

We thank the reviewer for their consideration. We agree and have added the data as Supplementary figure 6, reference in the Methods section line 501.

280: After this period?

We are referring to the period before travel restrictions were in place, so we've have clarified line 280 to "*for the period prior to travel restrictions*".

Figure 4 label: Should it be II instead of the EII?

The reviewer is correct, and we've corrected line 306.

DTW: is the DTW affected by different absolute values for the II?

Yes, which is why we normalized the data prior to running the analyses as we're just trying to minimize a distance measure(Tang and Müller 2009)

Tang, Rong, and Hans-Georg Müller. 2009. "Time-Synchronized Clustering of Gene Expression Trajectories." *Biostatistics* 10 (1): 32–45.